# Certified Robustness in Federated Learning

**Motasem Alfarra**    **Juan C. Pérez**    **Egor Shulgin**    **Peter Richtárik**    **Bernard Ghanem**

King Abdullah Univerity of Science and Technology (KAUST)
motasem.alfarra@kaust.edu.sa

## Abstract

Federated learning has recently gained significant attention and popularity due to its effectiveness in training machine learning models on distributed data privately. However, as in the single-node supervised learning setup, models trained in federated learning suffer from vulnerability to imperceptible input transformations known as adversarial attacks, questioning their deployment in security-related applications. In this work, we study the interplay between federated training, personalization, and certified robustness. In particular, we deploy randomized smoothing, a widely-used and scalable certification method, to certify deep networks trained on a federated setup against input perturbations and transformations. We find that the simple federated averaging technique is effective in building not only more accurate, but also more certifiably-robust models, compared to training solely on local data. We further analyze personalization, a popular technique in federated training that increases the model's bias towards local data, on robustness. We show several advantages of personalization over both (that is, only training on local data and federated training) in building more robust models with faster training. Finally, we explore the robustness of mixtures of global and local (*i.e.* personalized) models, and find that the robustness of local models degrades as they diverge from the global model. Our implementation can be found here.

## 1 Introduction

Machine learning models in general, and deep neural networks (DNNs) in particular, have demonstrated remarkable performance in several domains, including computer vision [1] and natural language processing [2], among others [3]. Despite this success, DNNs are brittle against small carefully-crafted perturbations in their input space [4, 5]. That is, a DNN $f_\theta : \mathbb{R}^d \to \mathcal{P}(\mathcal{Y})$ can produce different predictions for the inputs $x \in \mathbb{R}^d$ and $x + \delta$, although the adversarial perturbation $\delta$ is small, and thus $x$ and $x + \delta$ are indistinguishable to the human eye. Moreover, DNNs are also susceptible to input transformations that preserve an image's semantic information, such as rotation, translation, and scaling [6]. This observation raises security concerns about deploying such models in security-critical applications, *i.e.* self-driving cars. Consequently, this phenomenon sparked substantial research efforts towards building machine learning models that are not only accurate, but also robust. That is, building models that both correctly classify an input $x$, and maintain their prediction as long $x$ preserves its semantic content. Further interest arose in theoretically characterizing the output of models whose input was subjected to perturbations. Whenever a theoretical guarantee exists about a model's robustness for classifying inputs, $f_\theta$ is said to be *certifiably* robust [7]. Among several methods that built such certifiably robust models, Randomized Smoothing (RS) [8] is arguably one of the most effective, scalable, and popular approaches to certify DNNs. In a nutshell, RS predicts the most probable class when the classifier's input is subjected to additive Gaussian noise. While RS directly operated on pixel-intensity perturbations, a later work extended this technique to certify against input deformations, such as rotation and translation [9].

Workshop on Federated Learning: Recent Advances and New Challenges, in Conjunction with NeurIPS 2022 (FL-NeurIPS'22). This workshop does not have official proceedings and this paper is non-archival..

The problem of training large-scale machine learning models in the real world may need to account for data privacy constraints that arise in distributed setups. Federated learning (FL) [10, 11, 12] is a viable solution to this problem, which is now commonly used in popular services [13]. Although the certified robustness of models in traditional learning settings has been widely studied, few works consider the interaction between certified robustness and the federated setup. Specifically, federated learning distributes the training process across a (possibly large) set of clients (each with their own local data) and privately builds a global model without leaking nor sharing local raw data. In the federated learning setup, Zizzo *et al.* [14] showed that (1) adversarial attacks can reduce the accuracy of state-of-the-art federated learning models to 0%, and (2) adversarial robustness is higher in the centralized setting than in the federated setting, illustrating the difficulty of enhancing adversarial robustness. We identify two key and challenging settings for analyzing certified robustness in federated learning: **(i)** each client has insufficient data to fully train a robust model, and **(ii)** each client may be interested in building a personalized model that is robust against different transformations.

In this work, we set to study the certified robustness of models trained in a federated fashion. We deploy Randomized Smoothing to measure certified robustness against both pixel-intensity perturbations and input deformations. We first analyze when federated training benefits model robustness. We show that in the few local-data regimes—where clients have insufficient data—federated averaging improves not only model performance but also its robustness. Furthermore, we explore how certified robustness gains from personalization [15, 16, 17, 18], *i.e.* fine-tuning the global model on local data. We show that personalization can provide significant robustness improvements even when federated averaging operates on models trained without augmentations aimed at enhancing robustness. At last, we analyze a version of the recently-proposed federated mixture of global and personalized models [19, 20] from a certified robustness lens.

## 2 Related Work

**Certified Adversarial Robustness.** The seminal work of Szegedy *et al.* [4] demonstrated how DNNs are vulnerable to small perturbations in their input, now called adversarial attacks. Subsequent works noted how the brittleness against these attacks was widespread and easily exploitable [5, 21]. This observation led to further works on defense mechanisms that improved adversarial robustness [22, 23], and on attacks that aimed at breaking these mechanisms [21, 24]. To this aim, Cohen *et al.* [8] proposed Randomized Smoothing (RS), an approach for certification that proved to successfully scale with models and datasets. The original RS formulation certified models against pixel-level additive perturbations, and was studied in the federated learning setup by Chen *et al.* [25]. Recently, DeformRS [9] reformulated RS to consider parameterized deformations, granting robustness certificates against more general and semantic perturbations such as rotation.

**Federated Learning.** Federated Learning (FL) is a data-decentralized machine learning setting in which a central server coordinates a large number of clients (*e.g.* mobile devices or whole organizations) to collaboratively train a model [10, 11, 12, 26]. In particular, FL has been successfully deployed by Google [27], Apple [28], NVIDIA [29], and many others, even in safety- and security-critical settings, which motivates our work. Given how the inherent multiple-devices nature of FL introduces heterogeneity in the data, interest spurred towards techniques that considered model personalization [19, 20, 30, 31] to improve performance on each device's data [16, 18, 32, 33].

**Robustness in FL.** Concerns about the adversarial vulnerability of machine learning models have also been studied in the FL setup [34]. While most prior works focused on training-time adversarial attacks such as Byzantine attacks [35], and model [34] or data poisoning [36, 37], recent works studied test-time robustness. In particular, the works of Zizzo *et al.* [14] and Shah *et al.* [38] studied the incorporation of adversarial training [23], arguably the most effective empirical defense against adversarial attacks, into the FL setup. Closest to our study, is the recent work of Chen *et al.* [25] that explored certification via RS in FL. In contrast to [25], instead of pixel-level perturbations, we study realistic (*e.g.* rotation and translation) perturbations.

## 3 Methodology

**Notation.** We consider the standard image classification problem where a classifier $f_\theta : \mathbb{R}^d \to \mathcal{P}(\mathcal{Y})$, *e.g.* a DNN with parameters $\theta$, maps the input $x \in \mathbb{R}^d$ to the probability simplex $\mathcal{P}(\mathcal{Y})$ over $k$ classes, where $\mathcal{Y} = \{1, 2, \ldots, k\}$. That is, $\sum_{i=1}^{k} f_\theta^i(x) = 1$ with $f_\theta^i(x) \geq 0$ being the $i^{\text{th}}$ element of $f_\theta(x)$, representing the probability $x$ belongs to the $i^{\text{th}}$ class.

**Randomized Smoothing.** Randomized smoothing constructs a new "smooth" classifier from an arbitrary classifier $f_\theta$. In a nutshell, when this smooth classifier is queried at input $x$, it returns the average prediction of $f_\theta$ when its input $x$ is subjected to isotropic additive Gaussian noise, that is:

$$g(x) = \mathbb{E}_{\epsilon \sim \mathcal{N}(0, \sigma^2 I)} \left[ f_\theta(x + \epsilon) \right]. \tag{1}$$

Let $g$ predict label $c_A$ for input $x$ with some confidence, *i.e.* $\mathbb{E}_\epsilon[f_\theta^{c_A}(x + \epsilon)] = p_A \geq p_B = \max_{c \neq c_A} \mathbb{E}_\epsilon[f_\theta^c(x + \epsilon)]$, then, as proved in [39], $g_\theta$'s prediction is certifiably robust with radius:

$$R = \frac{\sigma}{2} \left( \Phi^{-1}(p_A) - \Phi^{-1}(p_B) \right), \tag{2}$$

where $\Phi^{-1}$ is the inverse CDF of the standard Gaussian distribution. That is, $\arg\max_i g^i(x + \delta) = \arg\max_i g^i(x)$, $\forall \|\delta\|_2 \leq R$. While the original RS framework certified against additive pixel perturbations in images, DeformRS [9] extended RS to certify against image *deformations*. DeformRS achieved this objective by proposing a *parametric-domain smooth classifier*. In particular, given an image $x$ with pixel coordinates $p$, a parametric deformation function $\nu_\phi$ with parameters $\phi$ (*e.g.* if $\nu$ is a rotation, then $\phi$ is the angle of rotation) and an interpolation function $I_T$, DeformRS defined the parametric-domain smooth classifier as:

$$g_\phi(x, p) = \mathbb{E}_{\epsilon \sim \mathcal{D}} \left[ f_\theta(I_T(x, p + \nu_{\phi + \epsilon})) \right]. \tag{3}$$

In a nutshell, $g$ outputs the average prediction of $f_\theta$ over transformed versions of the input $x$. Note that this formulation perturbs the pixels' *location*, rather than their intensities. DeformRS showed that, analogous to the smoothed classifiers in RS, parametric-domain smooth classifiers are certifiably robust against perturbations to the parameters of the deformation function with radius

$$
\begin{aligned}
\|\delta\|_1 &\leq \sigma \left( p_A - p_B \right) && \text{for } \mathcal{D} = \mathcal{U}[-\sigma, \sigma], \\
\|\delta\|_2 &\leq \frac{\sigma}{2} \left( \Phi^{-1}(p_A) - \Phi^{-1}(p_B) \right) && \text{for } \mathcal{D} = \mathcal{N}(0, \sigma^2 I).
\end{aligned}
\tag{4}
$$

Eq (4) states that, as long as the parameters of the deformation function (*e.g.* rotation angle) are perturbed by a quantity upper bounded by the certified radius, $g$'s prediction will remain constant.

**Specialization to Federated Training.** Earlier works proposed various training frameworks to enhance the robustness of smooth classifiers. The schemes studied by these frameworks ranged from simple data augmentation [8] to more sophisticated techniques, including adversarial training [40] and regularization [39]. In this work, we robustify the classifier via simple and nondemanding data augmentation. We select this approach since our main aim is to analyze the effect of federated training on certified robustness in isolation of additional and more sophisticated training schemes. In addition, this setup allows compatibility with other FL-specific constraints and technologies, such as differential privacy and secure aggregation, which can be essential for real-world deployment [13].

As in the Federated Learning setup, we employ the common federated averaging [10, 41] technique. In a nutshell, we consider the case when a dataset is distributed across multiple clients in a non-overlapping manner. We assume that all clients share the architecture $f_\theta$, to allow for weight averaging across clients. Furthermore, we study three different training schemes, where each client is minimizing the following regularized empirical risk [19]:

$$\min_{\theta_1, \dots, \theta_n} \frac{1}{n} \sum_{i=1}^n \lambda \mathcal{L}_i(\theta_i) + (1 - \lambda) \left\| \theta_i - \bar{\theta} \right\|_2^2, \tag{5}$$

where $\mathcal{L}_i$ is a training loss function (*e.g.* cross entropy) evaluated on local data of client $i$, $\theta_i$ represents the $i^{\text{th}}$ client's model parameters, $\lambda \in [0, 1]$ is a personalization parameter, and $\bar{\theta} = \frac{1}{n} \sum_{i=1}^n \theta_i$ is the average model.

**Local Training.** Each client trains solely on their own local data, without communication with other clients. At test time, each client employs their own local model. This setting is equivalent to minimizing the risk in Eq. (5) with $\lambda = 1$.

**Federated Training.** All clients are collaborating in building one global model. This is achieved by iteratively performing (1) client-side model updates with optimization on local data (*i.e.* solving Eq. (5) with $\lambda = 1$), and (2) communication of client models to the server, which averages all models and broadcasts the result back to the clients (*i.e.* solving Eq. (5) with $\lambda = 0$). This process is

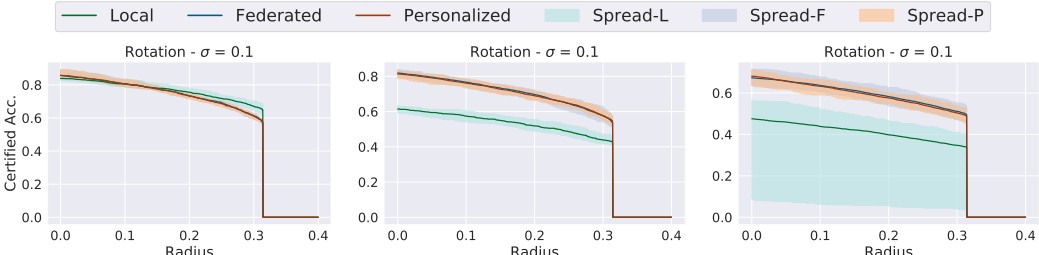

Figure 1: **Certified Accuracy against rotation.** We plot the certified accuracy curves for rotation deformation with $\sigma = 0.1$. We vary the number of clients on which we divide the dataset. Left: 4 clients, middle: 10 clients, right: 20 clients. As the number of clients increases, performance of local training worsens.

repeated until convergence or for a fixed number of communication rounds. At test time, all clients employ the global model sent from the server at the last communication round.

**Personalized Training.** Since [42] exposed how the basic FL formulation with $\lambda = 1$ and $\theta_1 = \cdots = \theta_n$ degrades the performance of some clients, we explore a personalization approach. Namely, upon federated training convergence, each client fine-tunes the global model communicated in the last round with a few optimization steps on their local data (*i.e.* solving Eq. (5) with $\lambda = 1$).

**Mixture of models.** We also consider a more sophisticated *personalized* ERM setting via a mixture of global and local models [19], seeking an explicit trade-off between the global model and the local models for $\lambda \in (0, 1)$. Our implementation for solving problem (5) for every client $i$ requires local gradient descent steps (with step size $\gamma$) and a mixing of current global and local models through:

$$\theta_i \leftarrow \theta_i - \gamma \left( \lambda \nabla \mathcal{L}_i(\theta_i) + 2(1 - \lambda)(\theta_i - \bar{\theta}) \right), \tag{6}$$

alternating with communication to the server for computing the current average model $\bar{\theta}$. At test time, each client uses its corresponding personalized model $\theta_i$. While the effect of federated training and personalization in improving model performance is widely explored in the literature, its impact on the trained model's certified robustness remains unclear.

## 4 Experiments

**Training Setup.** We fix ResNet18 [43] as our architecture and train on the CIFAR10 [44] and MNIST [45] datasets. We split the training and test sets across clients in a random and non-overlapping fashion. Note that this split forms a best-case condition for local training, allowing compatibility with federated training and its associated data-privacy constraints. We conduct 90 epochs of training, divided in 45 communication rounds, *i.e.* 2 epochs of local training per communication round. We set a batch size to 64, and use a starting learning rate of 0.1, which is decreased by a factor of 10 every 30 epochs (15 communication rounds). Unless specified otherwise, given a transformation we are interested in robustifying against (*e.g.* rotation), we train the model by augmenting the data with such transformation. This procedure is equivalent to estimating the output of the smooth classifier in Eq. (3) with one sample. We set the $\sigma \in \{0.1, 0.5\}$ for rotation and translation and $\sigma \in \{0.12, 0, 5\}$ for pixel perturbations, following [8, 9]. For local models, we train each client on their local data only with the aforementioned setup without any communication.

**Certification Setup.** For certification, we use the public implementation from [8] for pixel-intensity perturbations, and the one from [9] for rotation and translation. We use 100 Monte Carlo samples for computing the top prediction $c_A$, and 100k samples for estimating a lower bound to the prediction probability $p_A$; we further use a failure probability of $\alpha = 0.001$, and bound the runner-up prediction via $p_B = 1 - p_A$. For experiments on deformations, we choose $I_T$ to be a bi-linear interpolation function, following [9, 46]. To evaluate each client's certification, we fix a random subset of 500 local samples. At last, we plot the average certified accuracy curves across clients, highlighting the minimum and maximum values as the limits of shaded regions. We report "certified accuracy at radius $R$", defined as the percentage of test samples that is both, classified correctly, and has a certified radius of at least $R$. We report the certified accuracy curves for all clients in the appendix.

**Is Federated Training Good for Robustness?** We first investigate the benefits of federated training on certified robustness. To that regard, we use a rotation transformation and vary the number of

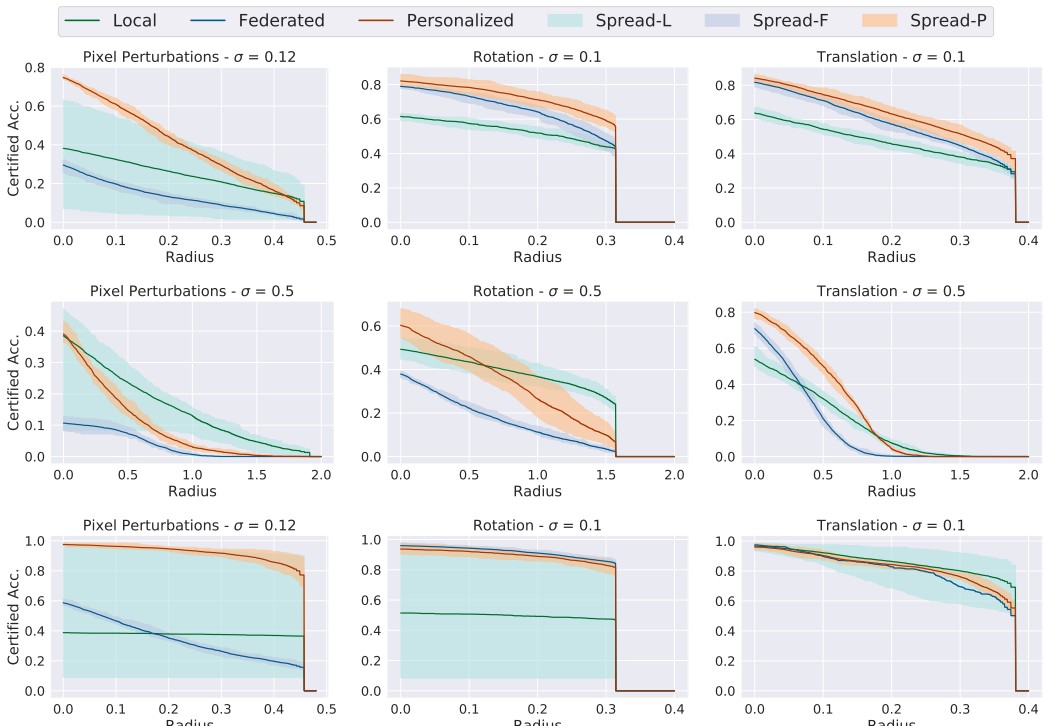

Figure 2: **Comparison between personalization and local training.** We plot the certified accuracy curves against pixel perturbations (left), rotation (middle), and translation (right) with varying $\sigma \in \{0.1, 0.5\}$ for rotation and translation an $\sigma \in \{0.12, 0.5\}$ for pixel perturbations. Last row: Experiments on MNIST dataset.

clients on which the dataset is distributed. We conduct experiments with 4, 10, and 20 clients, and report the results in Figure 1. From these results, we draw the following observations. **(i)** When the number of clients is small (4 clients), *i.e.* when each client has large data, local training can be sufficient for building a model that is accurate *and* robust. **(ii)** As the number of clients increases, *i.e.* a typical scenario in FL, and the amount of data available to each client is not large, federated averaging shines in providing a model with better performance and robustness. In particular, we find federated training can bring robustness improvements of over 20% in the few-local-data regime (10 or 20 clients). This can be interpreted as follows: when local data can sufficiently approximate the real-data distribution, then there is no advantage to conducting federated training. On the other hand, when local data is insufficient for approximating the data-distribution, federated learning is useful. **(iii)** The performance and robustness of personalized (fine-tuned) models are comparable to the global federated model. We attribute this observation to the federated model being trained with data augmentation, and thus converging to a reasonably-good local optimum. Hence, personalizing this model with few epochs of local training would not have very significant impact. We delve more into the benefits of personalization in the next section. **(iv)** At last, we observe that, as the number of clients increases, the performance difference between the best and worst performing clients is significantly larger for local training. This is evidenced by the spread in Figure 1, where the shaded area is notably larger for local training. We elaborate more on this phenomenon in the appendix.

**Is Personalization Beneficial for Robustness?** Next, we investigate the effect of personalization on certified robustness. In essence, the experiments in Figure 1 assumed all clients train robustly, aiming at collaboratively building a global robust model. However, in a more realistic federated setup, this is not necessarily the case. For instance, different clients could target robustness against different transformations or, in the extreme case, some clients could disregard robustness entirely and simply target accuracy.

To study this case, we use 10 clients and modify the averaging and personalization phases as follows. During the federated training phase, all models train without augmentations (*i.e.* targeting accuracy and disregarding robustness). During the personalization phase, we augment data with the transformation for which the client is interested in being robust against. Experimentally, we personalize

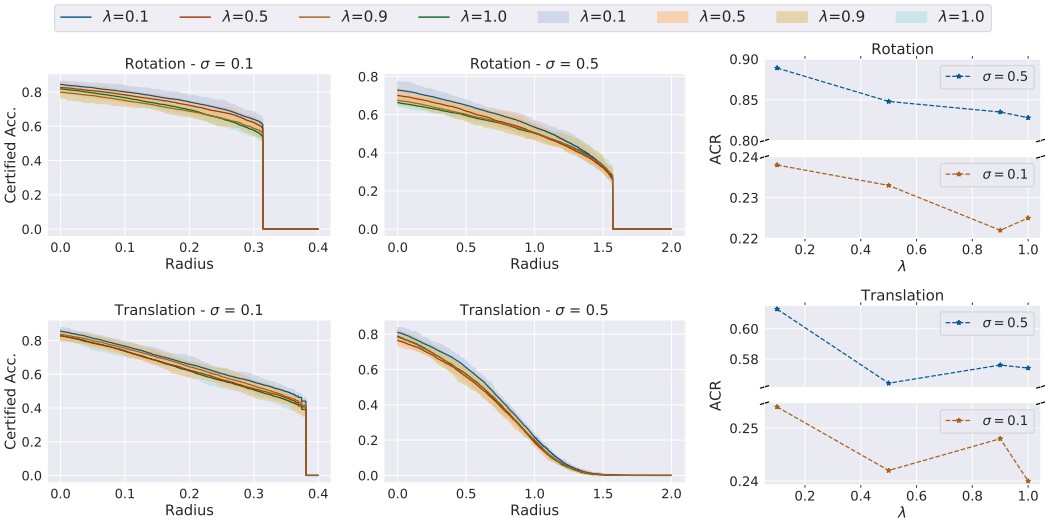

Figure 3: **Mixture of models.** We explore the effect of "intermediate" levels of personalization by varying $\lambda \in \{0.1, 0.5, 0.9, 1.0\}$ from Eq. (5). We experiment with rotation (top row) and translation (bottom row). The first two columns report certified accuracy curves, while the third one shows the associated the Average Certified Radius (ACR), *i.e.* area under the curves, as a function of $\lambda$. Our results show that the robustness of local models degrades, when they diverge from the global model.

against pixel perturbations, rotations, and translations, and report results in Figure 2. Our results show that personalization has a significant impact on improving model robustness. Moreover, this improvement is observed across the board, irrespective of the choice of the augmentation or $\sigma$.

**Advantages of Personalization.** Further, we address the following question: when is it better to robustly train on local data than to do robust personalization (starting from a nominally trained federated model)? We examine this question by comparing the performance of a nominally trained federated model that is robustly personalized (Figure 2) against local robust training. We display this comparison in Figure 2. We observe the following: **(i)** When the target local transformation is similar to the target federated one (small values of $\sigma$), personalized models significantly outperform local models, despite robust training only occurring in the personalization phase. That is, with far less computation and much faster training, federated training with personalization offers a better alternative to training only on local data. Furthermore, the performance gap increases with the number of clients—equivalent to assigning each client fewer local data. **(ii)** Following intuition, as the difference between the transformation in the federated and personalized settings increases, the performance of personalized models becomes comparable to that of their local counterparts. Note that this is an artifact of choosing a relatively small number of clients, on which the dataset is distributed. Indeed, in the appendix we report how a larger number of clients ($N = 100$) displays significant robustness improvements of personalized models compared to local models.

**FL with Mixture Model.** At last, we explore how using the mixture of models formulation in Eq. (5) affects robustness of the resulting local models. Note that through all the aforementioned experiments, we studied the case when $\lambda = 1$ for local training, or $\lambda = 0$ for federated training. To this end, we compare the performance of models trained with various values $\lambda \in (0, 1)$, leading to different personalization levels. That is, during local training, each client is leveraging the update step in (6) before communicating the local model to the orchestrating server for an averaging step.

Figure 3 plots the results of this experiment. The first two columns display certified accuracy curves against rotation and translation for different $\lambda$ values. The last column summarizes the results in terms of Average Certified Radius (ACR), *i.e.* the average certified radius of correctly-classified examples [39]. From these results we observe that: **(i)** for every deformation (rotation, translation) and every $\sigma$ value, the mixture of models formulation outperforms the robustness of standard federated training. This phenomenon is consistent with almost *any* value of the personalization parameter $\lambda$. In particular, we observe that setting $\lambda = 0.1$, corresponding to the *least* personalized local models, yields the best results. **(ii)** By contrasting these results to Figure 2, we notice that, across clients, the spread of certified accuracy is smaller than both naïve personalization and local training.

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

# Certified Robustness in Federated Learning
## Supplementary Materials

## A Additional Experiments

### A.1 Is Federated Averaging Beneficial for Robustness

In Section 4, we analyzed the benefits of personalization on the certified robustness when all clients are training nominally. Next, we address the following question: what if all other clients aim to build a model robust against transformations that are different than the desired one? We analyze the scenario when all clients train with an augmentation that differs from the one a certain client desires. Thus, we deploy augmentation with the general affine deformation [9] during the federated training phase, and then personalize with either rotation or translation. We report the results in Figure 4 and illustrate that personalization has a positive impact on robustness even when federated training is conducted with a different augmentation (an affine deformation, in this case).

In Figure 1, we analyzed the effect of varying the number of clients on the certified robustness against rotations. For completeness, we repeat the experiment with translations, and report the results in Figure 5. In particular, in this experiment we distribute CIFAR10 on 4 and 10 clients, and vary $\sigma \in \{0.1, 0.5\}$. Analogous to our earlier observation, as the number of clients increases, the local data becomes insufficient, and the performance and robustness of local training deteriorates.

### A.2 100 Clients Experiment

In Section 4, we observed that when the augmentation deployed in the federated training phase differs from the one used in the personalization phase, the performance of local models is comparable to that of personalized models. We scale our experiments by distributing the dataset on a larger number of clients (100). This makes the local data for each client insufficient for training local models. We plot the certified accuracy curves of local training, federated training, and personalized training in Figure 6. We observe that the performance gap increases between the personalized and local models, providing further evidence to support our hypothesis from Section 4. That is, federated training and personalization, compared to local training, provide a more reliable approach that improves both performance and robustness in more realistic federated scenarios.

### A.3 Ablation for All Clients

In all of our experiments, we reported the certified accuracy curves averaged across all clients, highlighting the minimum and the maximum performing clients in a shaded region. For completeness, we plot the certified accuracy for all clients for some of our experiments. In particular, for experiments with 4 clients, we show our results in Figure 7, and show the results on 10 clients in Figure 8. On each plot, we show the performance variation across clients when local training is conducted (compared to personalized training). We observe, following our earlier observations, that the spread is significantly higher for local training. Note that this behaviour is despite the fact that we distributed the data in a uniform and homogeneous fashion across clients. Hence, in more realistic scenarios with heterogeneous splits, this spread could grow, highlighting the reliability of federated

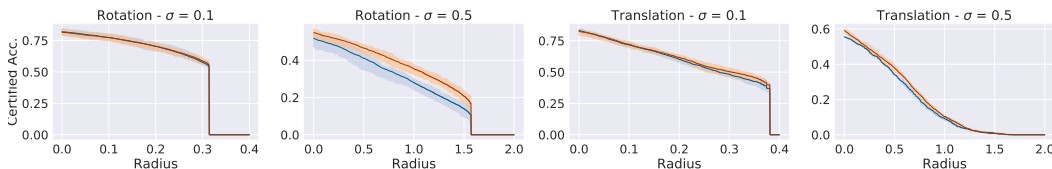

Figure 4: **Certified Robustness against Rotation starting from Affine. Effect of Personalization** Left: personalization with rotation augmentation with $\sigma \in \{0.1, 0.5\}$. Right: translation augmentation with $\sigma \in \{0.1, 0.5\}$. The color convention follows that of Figure 2.

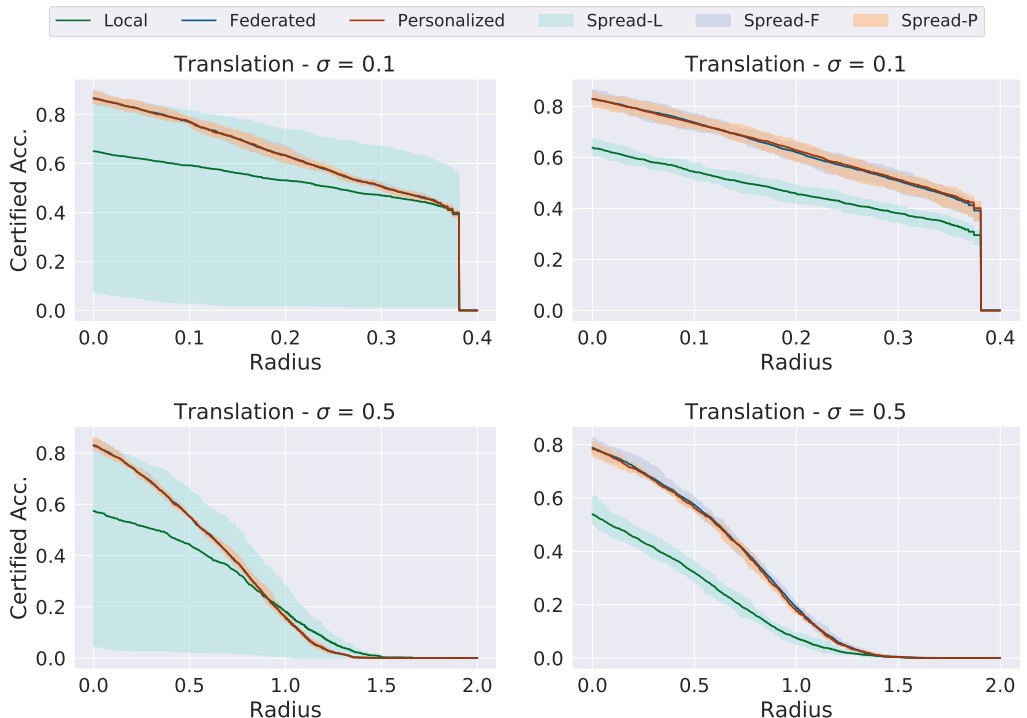

Figure 5: **Certified Accuracy against translation.** We plot the certified accuracy curves for translation deformation by varying $\sigma \in \{0.1, 0.5\}$ in the top and bottom rows, respectively. We vary the number of clients on which we divide the dataset. Left: 4 clients, right: 10 clients. As the number of clients increases, performance of local training worsens.

training and personalization in building accurate and robust models. We believe this analysis is an interesting setup for future work.

### A.4  Experimental Details

The experiments reported in the paper used the MNIST and CIFAR-10 [44] Datasets[1]. We used the available splits provided in PyTorch for training and testing sets. Moreover, we adopted the publicly-available codes for certification, as noted in the experimental section.

### A.5  FL with Mixture Model

At last, we experiment with the effect of the mixture model on the certified accuracy of the trained model. In Section 4, our experiments analyzed rotation and translation deformations on CIFAR10. For completeness, in Figure 9 we also report results against (1) pixel perturbations on CIFAR10 and (2) rotation, translation, and pixel perturbations on MNIST. Our results further confirm our observation in Section 4: the mixture model provides consistent improvement on the certified robustness for most considered values of $\lambda$. It is also worth noting that different degrees of personalization perform the best for different perturbations and $\sigma$ values.

## B  Social Impact

This paper explores (1) certified robustness and (2) federated learning. Both topics are associated to how machine learning settings relate to security which, in turn, has a social impact. In particular, robustness is associated with the secure deployment of models, hindering how malicious agents may attack recognition systems to fit their purposes. Furthermore, federated learning is a setting

---

[1]Available `https://www.cs.toronto.edu/ kriz/cifar.html`, under an MIT license.

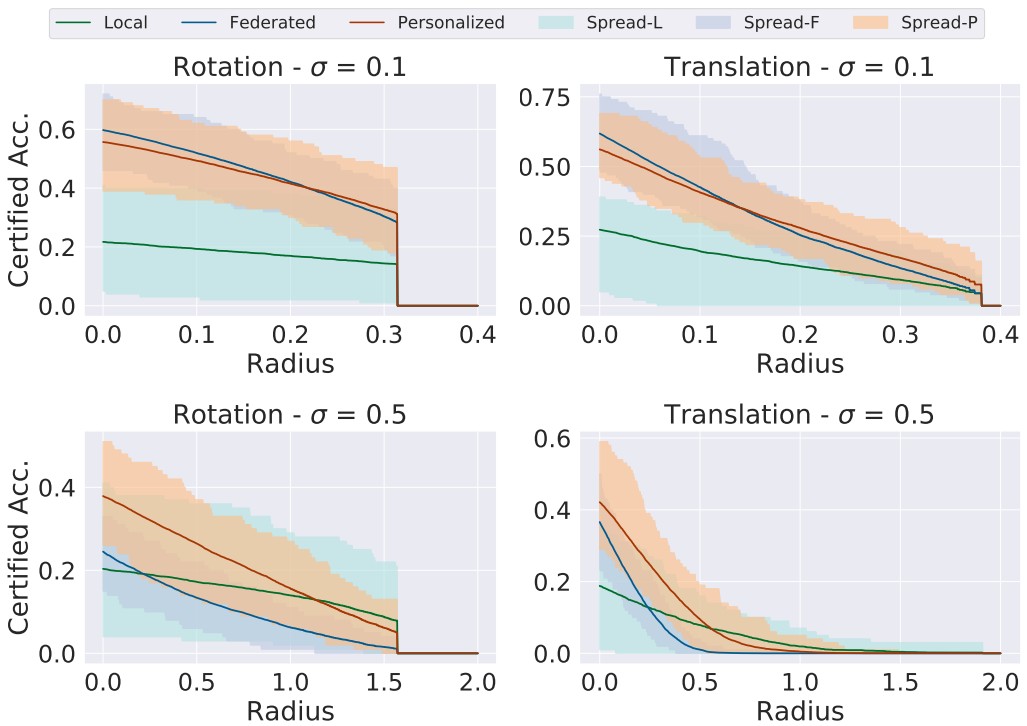

Figure 6: **Nominal pre-training and robust personalization with 100 clients.** We compare local robust training to nominal federated training with robust personalization in terms of certified accuracy on CIFAR10.

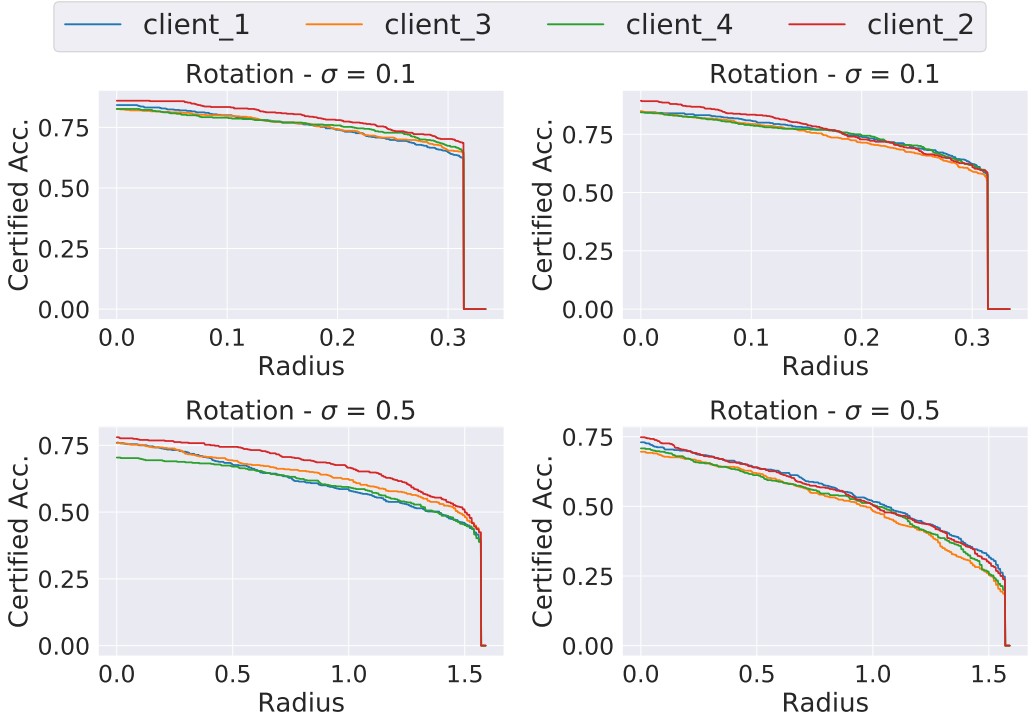

Figure 7: **Ablations for all clients. 4 Clients experiment** Left: Local training. Right: Personalized training.

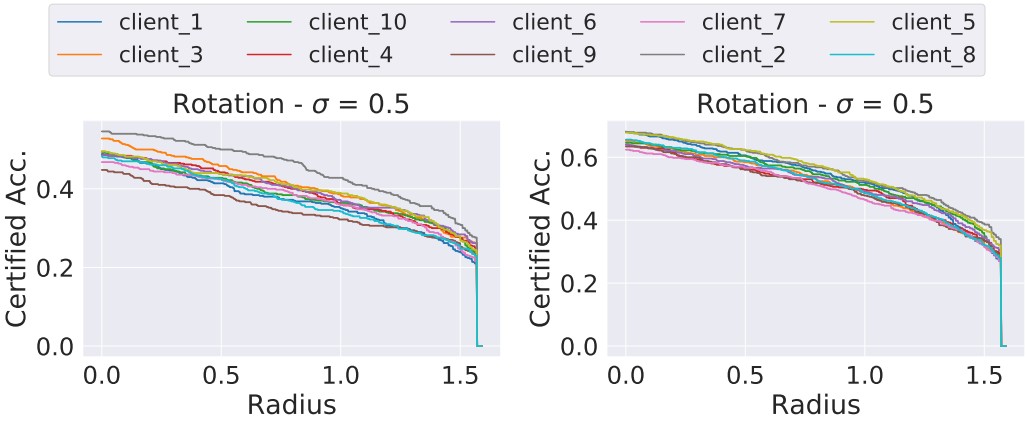

Figure 8: **Ablations for all clients. 10 Clients experiment** Left: Local training. Right: Personalized training.

that arises naturally when data-privacy constraints are introduced into the learning objective: how to develop well-performing models that leverage user data while preserving privacy.

## C  Compute Power

All in all, our work has the potential to be used by services targeting high-performance private models to benefit users. In particular, we used 20 Nvidia-V100 GPUs for all of our experiments.

## D  Limitations

There exist several frameworks that characterize federated training and personalization. A limitation of our work is the focus on the popular federated averaging technique. A possible interesting avenue for future work is to benchmark different federated and personalization frameworks in terms of robustness. Moreover, new mixture models could be devised that explicitly target robustness enhancement are left for future works. Moreover, the development of new mixture models explicitly aimed at improving robustness and reliability is also left for future work. At last, in this work we considered a homogeneous split of data across clients. That is, all clients have the same amount, and probably the same set of classes, of data. We note here, while this setup is less realistic to the real world setup where different clients have heterogeneous data splits, we remark here that this serves as the best condition for local training. That is, even when putting local training at advantage, our work showed the superiority of conducting federated training and personalization in providing models that are both accurate and robust. We leave studying the heterogeneous setup for future works.

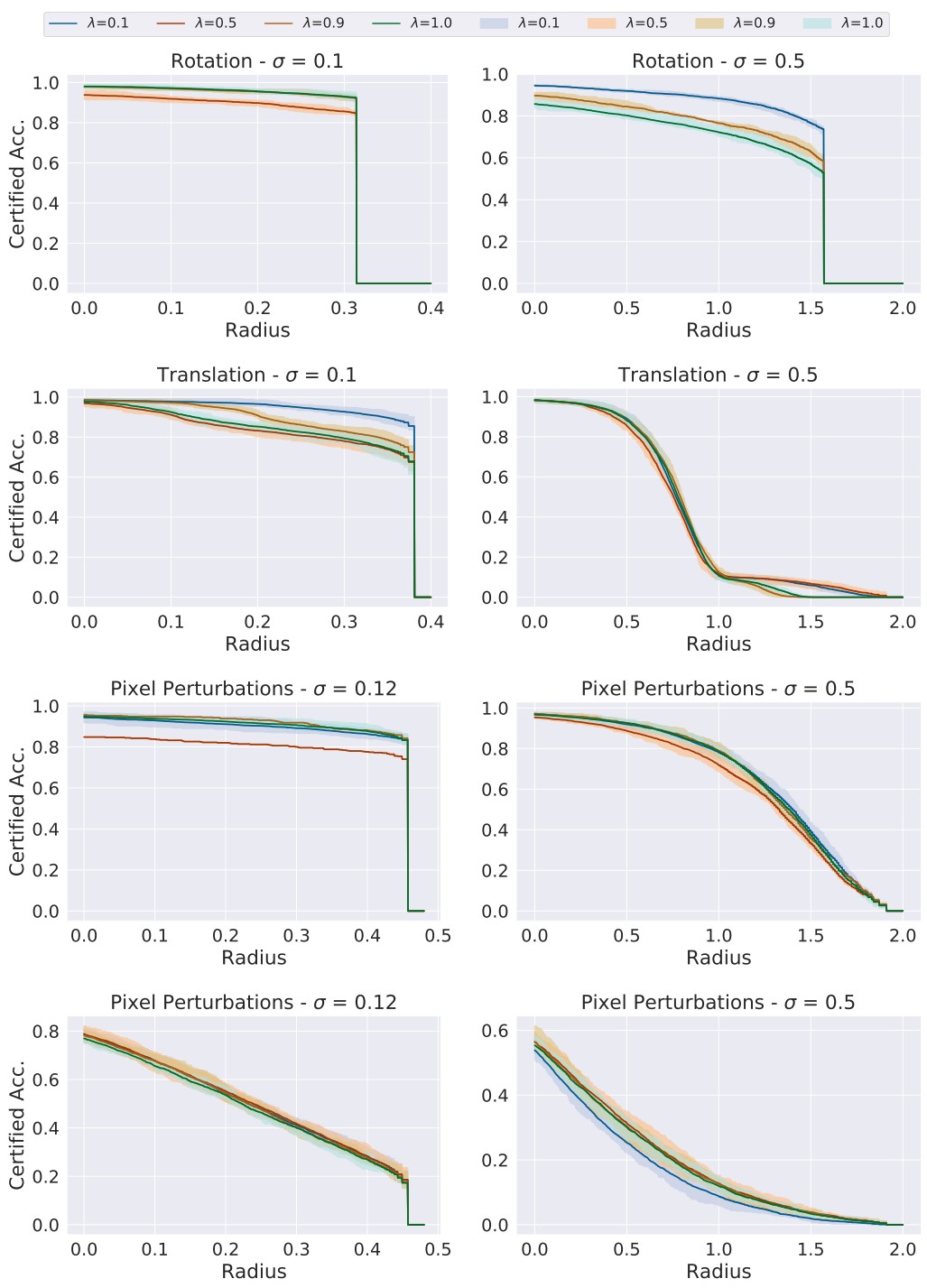

Figure 9: **FL with Mixture of Models.** We analyze the effect of mixture model on certified robustness on MNIST (first three rows) and CIFAR10 (last row). We observe that for most personalization values $\lambda$, the mixture of model outperforms the naive federated averaging model.

