# OpenReview forum: "Certified Robustness in Federated Learning"
_NeurIPS.cc/2022/Workshop/Federated_Learning — FL-NeurIPS 2022 Poster_

### Official Review · Reviewer_Jexy · 2022-10-14
**a borderline paper**

Strengths:
1. The interplay between federated training, personalization, and certified robustness is an interesting problem.
2. Overall structure is good.

This paper deploys randomized smoothing to certify deep networks trained on a federated setup against input perturbations and transformations. This paper finds that the simple federated averaging technique is effective in building not only more accurate, but also more certifiably-robust models, compared to training solely on local data. This paper further shows several advantages of personalization in building more robust models with faster training. Finally, this paper explores the robustness of mixtures of global and local (i.e. personalized) models, and finds that the robustness of local models degrades as they diverge from the global model.

Weaknesses:
1. the experiment setting is eather limited, only homogeneous setting is considered.
2. need to explain what is Spread-X in all figures.
3. the chosen datasets are too simply, only CIFAR10 and MNIST datasets are studied.
4. theory is always expected in certified robustness.
5. only fedavg was considered.
6. no conclusion was drawn.

typos: #49: We f identify

Comment:
1. the y-axis range should be consistent in Figure 1, to better illustrate "As the number of clients increases, performance of local training worsens".

---

### Official Review · Reviewer_k2XT · 2022-10-17

Summary

This paper explores the certified robustness in federated learning via randomized smoothing against pixel-intensity perturbation and input deformations. The authors empirically evaluate the certified robustness of three kinds of federated learning algorithms on CIFAR10 and MNIST: FedAvg, personalization via fine-tuning the global model, and a mixture of global and personalized models.



Strength
1. The evaluation presents interesting results that characterize the relationship between certified robustness and different FL setups (e.g., # clients, FL training algorithms).
2. The paper is well-written and easy to follow.


Weakness
1. Limited novelty: the authors do not propose any new algorithms throughout the paper and mainly focus on the evaluation of randomized smoothing under different FL algorithms, which are all existing techniques.
2. It would be great if the authors could discuss the existing work on certified robustness in FL [1,2]. In fact, in FL, training-time attacks are of unique interest/challenges because of the distributed learning methodology and the existence of malicious agents, while the test-time attacks mainly follow the well-established attacks in centralized learning (e.g.,pixel-intensity perturbation and input deformations as considered in the paper).
3. Evaluation of the empirical tightness of the bound under adversarial attacks might further strengthen the submission.
4. Typo: line 49: “We f identify”

References:

[1] Crfl: Certifiably robust federated learning against backdoor attacks, ICML,  2021.

[2] Provably secure federated learning against malicious clients. In Proceedings of the AAAI Conference on Artificial Intelligence, volume 35, pages 6885–6893, 2021.

---

### Decision · Program_Chairs · 2022-10-20

Accept (Poster)